# COMPAR-EU Recommendations on Self-Management Interventions in Type 2 Diabetes Mellitus

**DOI:** 10.3390/healthcare12040483

**Published:** 2024-02-16

**Authors:** Jessica Beltran, Claudia Valli, Melixa Medina-Aedo, Carlos Canelo-Aybar, Ena Niño de Guzmán, Yang Song, Carola Orrego, Marta Ballester, Rosa Suñol, Janneke Noordman, Monique Heijmans, Georgios Seitidis, Sofia Tsokani, Katerina-Maria Kontouli, Christos Christogiannis, Dimitris Mavridis, Gimon de Graaf, Oliver Groene, Maria G. Grammatikopoulou, Francisco Camalleres-Guillem, Lilisbeth Perestelo-Perez, Helen McGloin, Kirsty Winkley, Beate Sigrid Mueller, Zuleika Saz-Parkinson, Rosa Corcoy, Pablo Alonso-Coello

**Affiliations:** 1Iberoamerican Cochrane Centre, Sant Antoni Maria Claret, 167, 08025 Barcelona, Spain; beltranpuerta.jessica@gmail.com (J.B.); mmedinaa@santpau.cat (M.M.-A.); carlos.canelo.ay@gmail.com (C.C.-A.); ena.ninodeguzman@gmail.com (E.N.d.G.); songyang@cuhk.edu.cn (Y.S.); 2Institut de Recerca Sant Pau (IR SANT PAU), Sant Quintí 77-79, 08041 Barcelona, Spain; 3Avedis DonabedianResearch Institute (FAD), Universitat Autònoma de Barcelona, 008037 Barcelona, Spain; cvalli@fadq.org (C.V.); corrego@fadq.org (C.O.); mballester@fadq.org (M.B.); rsunol@fadq.org (R.S.); 4Network for Research on Chronicity, Primary Care, and Health Promotion (RICAPPS), 08007 Barcelona, Spain; 5Netherlands Institute for Health Services Research (NIVEL), 3513 CR Utrecht, The Netherlands; j.noordman@nivel.nl (J.N.); m.heymans@nivel.nl (M.H.); 6Department of Primary Education, School of Education, University of Ioannina, 45110 Ioannina, Greece; g.seitidis@uoi.gr (G.S.); s.tsokani@uoi.gr (S.T.); kmkontouli@uoi.gr (K.-M.K.); c.christogiannis@uoi.gr (C.C.); dmavridi@uoi.gr (D.M.); 7Institute for Medical Technology Assessment, Erasmus University Rotterdam, P.O. Box 1738, 3062 PA Rotterdam, The Netherlands; degraaf@imta.eur.nl; 8OptiMedis, 20095 Hamburg, Germany; o.groene@optimedis.de; 9Immunonutrition and Clinical Nutrition Unit, Department of Rheumatology and Clinical Immunology, Medical School, University of Thessaly, Biopolis Campus, 43100 Larissa, Greece; 10Spanish Society of Family Medicine, Infanta Mercedes Primary Care Centre, 28020 Madrid, Spain; fcamarelles@telefonica.net; 11Evaluation Unit (SESCS), Canary Islands Health Service (SCS), Network for Research on Chronicity, Primary Care, and Health Promotion (RICAPPS), 38109 Tenerife, Spain; lilisbeth.peresteloperez@sescs.es; 12School of Nursing, Health Science and Disability Studies, ATU St Angelas, F91 C643 Sligo, Ireland; hmcgloin@stangelas.nuigalway.ie; 13Florence Nightingale Faculty of Nursing, Midwifery & Palliative Care, King’s College London, London SE1 8WA, UK; kirsty.winkley@kcl.ac.uk; 14Institute of General Practice, Faculty of Medicine, University Hospital Cologne, University of Cologne, 50923 Cologne, Germany; beate.mueller@uk-koeln.de; 15European Research Executive Agency, 1000 Brussels, Belgium; zsazparkinson@gmail.com; 16CIBER Bioengineering, Biomaterials and Nanotechnology, Instituto de Salud Carlos III, 28220 Madrid, Spain; rcorcoy@santpau.cat; 17Centro de Investigación Biomédica en Red de Epidemiología y Salud Pública (CIBERESP), 28029 Madrid, Spain

**Keywords:** diabetes mellitus, type 2, self-management interventions, recommendations, GRADE approach

## Abstract

Self-management interventions (SMIs) offer a promising approach to actively engage patients in the management of their chronic diseases. Within the scope of the COMPAR-EU project, our goal is to provide evidence-based recommendations for the utilisation and implementation of SMIs in the care of adult individuals with type 2 diabetes mellitus (T2DM). A multidisciplinary panel of experts, utilising a core outcome set (COS), identified critical outcomes and established effect thresholds for each outcome. The panel formulated recommendations using the Grading of Recommendations, Assessment, Development, and Evaluations (GRADE) approach, a transparent and rigorous framework for developing and presenting the best available evidence for the formulation of recommendations. All recommendations are based on systematic reviews (SR) of the effects and of values and preferences, a contextual analysis, and a cost-effectiveness analysis. The COMPAR-EU panel is in favour of using SMIs rather than usual care (UC) alone (conditional, very low certainty of the evidence). Furthermore, the panel specifically is in favour of using ten selected SMIs, rather than UC alone (conditional, low certainty of the evidence), mostly encompassing education, self-monitoring, and behavioural techniques. The panel acknowledges that, for most SMIs, moderate resource requirements exist, and cost-effectiveness analyses do not distinctly favour either the SMI or UC. Additionally, it recognises that SMIs are likely to enhance equity, deeming them acceptable and feasible for implementation.

## 1. **Summary of Recommendations**

### 1.1. Recommendation for SMIs vs. UC

For patients with T2DM, the COMPAR-EU panel is in favour of using self-management interventions, rather than UC (conditional recommendation based on very low certainty in the evidence of effects ⨁◯◯◯).

**Remark:** SMIs should be of high intensity (more than ten hours of contact) and include education, monitoring, behavioural techniques (either action-based or emotional-based), and/or social support components. Patients with T2DM with poor glycaemic control (poor HbA1c at baseline) are likely to experience a larger effect.

### 1.2. Recommendations for selected SMI vs. UC

For patients with T2DM, the COMPAR-EU panel is in favour of using the following self-management interventions, rather than UC (conditional recommendation based on low certainty of the evidence about the effects ⨁⨁◯◯):-Monitoring techniques led by peers delivered in groups, with or without professional support;-Emotional-based behavioural techniques led by peers delivered remotely, with or without professional support;-Monitoring, action-based behavioural techniques, and shared decision making delivered in groups;-Monitoring, action-based and emotional-based behavioural techniques, and social support led by peers with or without professionals delivered remotely;-Emotional-based behavioural techniques and social support delivered in groups-Action-based behavioural techniques, social support led by peers and professionals;-Education delivered in groups and remotely;-Monitoring techniques and social support delivered remotely;-Monitoring and action-based behavioural techniques, shared decision making, and social support, delivered in groups;-Monitoring and emotional-based behavioural techniques delivered remotely.

### 1.3. What Do These Recommendations Mean?

Many people with T2DM may consider using SMIs as part of their treatment, but not everyone. Clinicians should recognise that different choices will be appropriate for different patients, assisting each patient in making a treatment decision that aligns with their own values and preferences.

## 2. Introduction

T2DM is one of the most prevalent chronic diseases worldwide, with a trend of continuous growth of incidence and prevalence rates. It is also a leading cause of disability, mortality, and reduced life expectancy [1]. Its associated health expenditure has rapidly grown from USD 232 billion in 2007 to USD 966 billion in 2021 and is expected to continue increasing [2]. As a chronic disease, T2DM requires continuous, patient-centred medical care and management to prevent disease progression, with active patient involvement in their own care. Therefore, self-management appears to be a promising approach to dealing with the challenges related with T2DM management [3].

Self-management (SM) is defined as “the person’s ability to control symptoms, medication, physical and psychosocial issues, and lifestyle changes required by living with a chronic illness to maintain an adequate quality of life” [4,5]. SMIs aim to increase patients’ (and caregivers’) skills and confidence to actively participate in the management of their condition [6,7]. SMIs such as education-based programmes have shown clinically relevant improvements in glycaemic control, T2DM knowledge, triglyceride levels, blood pressure, medication reduction, and weight management [8,9,10]. SMIs may also improve measures of psychological health, self-efficacy, health-related quality of life, and adherence [11,12,13,14,15,16,17]. However, despite these benefits, a low rate of implementation of SMIs has been reported, due to different barriers, such as logistical issues and lack of perceived benefit [18,19].

The aim of this document is to provide evidence-based clinical recommendations for the use and implementation of SMIs in adults with T2DM. These recommendations were developed using the Grading of Recommendations, Assessment, Development, and Evaluations (GRADE) approach, a recognised methodological framework esteemed for its transparent and rigorous evaluation of evidence to formulate recommendations [20]. The formulation of these recommendations is an integral part of the COMPAR-EU project, a multimethod, interdisciplinary project which aims to bridge the gap between current knowledge and practice of SMIs for adults in Europe living with one of four high-priority chronic conditions, including T2DM [21] (more information available at http://self-management.eu/, accessed on 24 January 2024).

## 3. Materials and Methods

This study was conducted as part of a larger study, COMPAR-EU (see Box 1), and uses the descriptive data on T2DM collected there.

Box 1The COMPAR-EU ProjectCOMPAR-EU was a multimethod, interdisciplinary project that contributed to bridging the gap between current knowledge and practice of self-management interventions. COMPAR-EU aimed to identify, compare, and rank the most effective and cost-effective self-management interventions (SMIs) for adults in Europe living with one of the four high-priority chronic conditions: type 2 diabetes, obesity, chronic obstructive pulmonary disease, and heart failure. The project provides support for policymakers, guideline developers and professionals to make informed decisions on the adoption of the most suitable self-management interventions through an IT platform, featuring decision-making tools adapted to the needs of a wide range of end users (including researchers, patients, and industry).COMPAR-EU launched in January 2018 and was completed in December 2022, contributing the following outputs: (i) an externally validated taxonomy composed of 132 components, classified in four domains (intervention characteristics, expected patient (or carer) self-management behaviours, type of outcomes, and target population characteristics), (ii) core outcome sets (COS) for each disease, including 16 outcomes for chronic obstructive pulmonary disease (COPD), 16 for Heart Failure, 13 for T2DM, and 15 for Obesity, (iii) extraction and descriptive results for each disease based on 698 studies for diabetes, 252 studies for COPD, 288 studies for heart failure and 517 studies for obesity, (iv) comparative effectiveness analysis based on a series of pairwise meta-analyses, network meta-analyses (NMAs), and component NMAs (cNMA) for all outcomes across all four diseases, (v) contextual analysis addressing information on equity, acceptability, and feasibility; general information on contextual factors on the level of patients, professionals, their interaction, and the health care organisation for those interested in implementation, (vi) cost effectiveness conceptual models have been created for each chronic condition, including risk factors or intermediate variables relevant for SMIs and final outcomes, and (vii) business plans and a sustainability strategy was developed based on a multipronged approach, including qualitative interviews with managers and clinicians, a focus group with clinical representatives from EU countries, workshops with industry representatives, and a hackathon event.The majority of the COMPAR-EU end-products are available on the online COMPAR-EU platform: www.self-management.eu (accessed on 24 January 2024). Watch the introductory video about the decision aids: https://www.youtube.com/watch?v=_nqy6s79ZcY (accessed on 24 January 2024). We used the GRADE approach to assess the certainty of the evidence and to formulate recommendations on the use and implementation of SMIs [22,23,24,25,26,27,28].Recommendations were formulated by a multidisciplinary panel that included health services researchers, endocrinologists, health economists, family medicine practitioners, self-management experts, nurses, nutritionists, clinical psychologists, patient advocates, statisticians, and methodologists. Panel members voluntarily joined the panel via an online open call. All panellists disclosed financial and non-financial interests (link to read more about panel members: https://self-management.eu/compar-eu-project/diabetes-panel/ (accessed on 24 January 2024)).

To inform the recommendations, the COMPAR-EU consortium partners conducted a systematic review (SR) with standard pairwise meta-analyses, NMA, and cNMA to assess SMIs effectiveness, a cost-effectiveness analysis, a scoping review and an overview [29,30] to inform about patients values and preferences, and a contextual analysis to inform about acceptability, feasibility, and other implementation considerations [21].

### 3.1. Structured Question, Outcome Prioritisation, and Decision Thresholds

Our research question was: *“Which are the most effective and cost-effective SMIs for adults with T2DM?”*. See below the PICO (Population, Intervention, Comparison, and Outcomes) format (see Table 1).

The population was defined as adult patients with T2DM, regardless of the severity of the condition, presence of comorbidities, or any other risk factor for developing complications. The population characteristics, along with other features of studies included in the SR, are detailed in a descriptive analysis published elsewhere [31]. Briefly, across all studies, participants were predominantly female (mean 57%, SD 49–67%), with a mean age of 58 years and an average time since diagnosis of 8.6 years. While most studies involved general samples of T2DM, 13% focused on T2DM patients with comorbidities, such as obesity, hypertension, and depression, among others.

SMIs are complex interventions composed of multiple components (such as support techniques, formats, target behaviours, type of providers delivering the intervention, etc.) that may interact with each other and/or with environmental factors (such as setting and health system characteristics, etc. [32]. This results in a large diversity of SMIs and makes it challenging to adequately group them when synthesising and measuring their effectiveness [33]. Therefore, we took into account this complexity by grouping the SMIs according to the reported support techniques, type recipient, type of provider, and delivery methods, using the definitions of the COMPAR-EU taxonomy [6] (see examples in Table 2 and the definition of components according to the COMPAR-EU taxonomy) (Appendix A).

For the recommendations, we presented results comparing SMIs with UC, which mainly included standard of care treatment, clinical visits, and information sharing. While other comparators were considered in the NMA, we selected UC as the reference intervention. This choice was based on its strong connectivity (most connected node) across outcomes and its practical significance [28].

We considered the outcomes included in the COMPAR-EU core outcome set (COS). This COS was based on an extensive literature search, a two-round Delphi process involving people living with T2DM and healthcare professionals and a final in-person consensus workshop (Complete COS for T2DM). Our SR included evidence for 24 outcomes (including different types of measures; for example, blood pressure was measured both as systolic blood pressure and as diastolic blood pressure). Following GRADE guidance, the panel agreed on the definitions of health outcomes and rated the importance of each outcome. The panel classified these 24 outcomes into those considered as critical for decision making (N = 13) and those considered important but not critical (N = 11), using an online survey (see Appendix A) [35]. The panel considered the critical outcomes to inform recommendations, while important outcome data were used as complementary evidence.

The panel also agreed on the magnitude of decision thresholds for all outcomes to determine the effect size (i.e., large effect, moderate effect, small effect, and trivial to no effect) [36]. The magnitude thresholds were based on a review of literature, whether they reported a value for each category or a minimal important difference (MID) value that was considered as a small effect, and then to define the remaining categories, we assumed that a large effect would be twice the MID. When the MID was not available in the literature for a given outcome, we searched for SRs of randomised controlled trials (RCTs) that reported an effect on the outcome consistent with an improvement in cardiovascular events or mortality. An effect estimate was considered as a small effect if correlated with a 20% reduction in cardiovascular events or mortality, and then to define the remaining categories, we also assumed that a large effect would be twice the MID. When the summary measure was a standardised mean difference (SMD), we used modified ‘Cohen’s effect sizes’ to determine these magnitude thresholds [37,38].

### 3.2. Systematic Review Evidence

#### 3.2.1. Effectiveness

In the SR, a total of 11,780 unique citations were retrieved from database searches, 10,125 were excluded during title and abstract screening, 1026 were appraised as full text, and 361 were finally excluded due to differences in population, outcomes of interest, and design, or were considered as multiple publications of the same RCT. The SR included 665 trials that compared a wide spectrum of SMIs in adult patients with T2DM (Appendix A).

We conducted three different analyses on the comparative effectiveness and safety of SMIs: standard pairwise MA, NMA, and CNMA, which is an extension of NMA to explore components’ effects separately [39,40,41]. In the pairwise MA, we compared all SMI vs. UC for all critical and important outcomes. Such an analysis would inform us whether SMIs work in general, irrespective of the components employed. We also conducted additional pairwise MA for long-term complications and mortality, comparing specific interventions vs. UC. We used the COMPAR-EU taxonomy to classify SMIs according to their components. Then, for each outcome, we conducted NMAs to explore which combinations of components work and, since we expected a sparse network with imprecise NMA effect estimates, we conducted a CNMA to explore the effectiveness of each one of the pre-specified components.

#### 3.2.2. Values and Preferences

We conducted a scoping review of reviews to summarise patients’ and caregivers’ experiences and preferences on SMIs, and to identify the most relevant outcomes from their perspective. Methods and results are published elsewhere [29]. Furthermore, we conducted an overview of quantitative and qualitative reviews exploring patients’ and informal caregivers’ values and preferences to analyse in-depth and describe how T2DM patients and caregivers value outcomes. For each critical outcome, quantitative evidence was summarised by pooling utility (or disutility) values (when feasible), and qualitative evidence was analysed following an inductive–deductive thematic synthesis [30]. The certainty of the evidence was evaluated following the GRADE principles [42].

#### 3.2.3. Resource Use and Cost-Effectiveness

We aimed to assess the cost-effectiveness of SMIs in T2DM. The UKPDS Outcomes Model 2 (UKPDS OM2) was used to estimate the long-term health effects of SMIs in T2DM [43]. The UKPDS OM2 makes use of a set of risk equations to translate effects on short term clinical outcomes (e.g., HbA1c, BMI, and blood pressure) into effects on microvascular and macrovascular complications and mortality. The short-term effects of SMIs (HbA1c, body mass index, high-density lipoprotein (HDL), low-density lipoprotein (LDL), and systolic blood pressure (SBP)) were derived from the NMA. The disease model was populated by country specific and disease specific data (when possible). The health economic models adopted a societal perspective. To incorporate the broader impact on society, the impact of T2DM complications on productivity and informal care use was estimated using the data from the SHARE database [44,45]. Future unrelated medical and nonmedical costs were estimated for each country of interest (methods and results reported elsewhere [46]).

Due to the large diversity of SMIs that were grouped in each node, it was not possible to define the resource requirements per node. Instead, a headroom analysis was conducted to estimate the maximum cost at which a given SMI could still be considered cost-effective. We used two threshold values: EUR 20,000 and EUR 50,000 per quality-adjusted life-year (QALY) considering that the willingness to pay may vary across European countries. To provide context for these results, a literature search was conducted to collect cost estimates of SMIs in T2DM. We also conducted a budget impact analysis for the SMIs for T2DM, by country, highlighting the financial consequences of implementing SMIs. To estimate the budget impact, the size of the T2DM population per country was multiplied by the maximum intervention costs of the SMI, at which the SMI is considered cost-effective at the given cost-effectiveness thresholds.

#### 3.2.4. Contextual Factors

A contextual framework was developed to describe the most relevant contextual factors for the implementation of SMI, and thus, to explain how SMIs work within the different contexts and for specific patient groups. We conducted a SR of the literature to identify potential contextual factors, and we structured them according to the seven domains and (sub) determinants of the comprehensive, integrated checklist of determinants of practice (TICD) framework [47,48]. We conducted a Delphi study in which stakeholders identified and prioritised the most important contextual factors for different components of SMIs [47]. Finally, we identified contextual factors for the selected SMIs through the review of the trials included in the NMA.

### 3.3. Certainty Assessment and Selection of the Interventions

We assessed the certainty of the evidence using the GRADE approach [35]. We assessed the certainty of the NMA evidence using the four-step NMA GRADE approach [24,25,26,27]. We also used the GRADE approach to assess the cost-effectiveness and resource use certainty assessment [49] and values and preferences evidence [42].

#### Prioritisation of Interventions

To formulate recommendations using the evidence of effects from such a complex set of NMAs, we considered it necessary to narrow down the number of interventions. For each outcome, we classified all the estimates of effect in a hierarchical order according to their magnitude of effect (point estimate). Then, we prioritised the interventions with the highest certainty of the evidence [28]. We checked that results were in agreement with the p-score values, a ranking metric assuming values from zero (worst) to one (best) [50]. Then, we selected SMIs following these criteria: SMIs with beneficial effects in at least two critical independent outcomes and with the highest certainty evidence. When an intervention was highly beneficial for only a single critical outcome and the certainty was higher than the certainty level of the SMIs showing beneficial effects in at least two critical independent outcomes, it was also selected.

### 3.4. Evidence to Recommendations

Using GRADE Evidence to Decision (EtD) frameworks, the recommendations were based on the assessment of the benefits and harms of the SMIs and the considerations of costs, patients’ values and preferences, acceptability, feasibility, and equity. Before our final guideline panel meeting, we asked each panellist to complete the judgements of each framework. The purpose of these frameworks is to help panellists use the evidence summaries in a structured and transparent way to develop the final recommendations. Four meetings were organised to review and discuss the evidence. The technical team formulated draft judgements and sent them to the panel for online voting using the Panel Voice GRADEpro extension. During the final meeting, the panel reviewed the voting results and considered the practical implications of those judgements for their recommendations. The panel formulated recommendations for the selected SMIs. However, given that the NMA was very sparse and imprecise, the panel also formulated a recommendation using the evidence from the standard pairwise meta-analysis (for the comparison “All SMIs vs. UC”) and from the CNMA.

## 4. Results

### 4.1. Recommendation for All SMI vs. UC

For patients with T2DM, the COMPAR-EU T2DM panel is in favour of using self-management interventions, rather than UC (conditional recommendation based on very low certainty in the evidence of effects ⨁◯◯◯).

**Remark:** SMIs should be of high intensity (more than ten contact hours) and include education and monitoring plus one or more additional components, such as behavioural techniques (either action-based or emotional-based) and/or social support. Patients with T2DM with poor glycaemic control (poor HbA1c at baseline) are likely to have a larger benefit than those with a well-controlled glycaemic level.

#### 4.1.1. Summary of the Evidence

#### Evidence of Effects

The standard meta-analysis included 335 trials comprising 58,621 patients. Analysis of RCTs showed that the majority of SMI have a trivial or marginal beneficial effect across outcomes, according to the magnitude’s thresholds defined a priori by the panel. Despite the benefits being considered not clinically relevant for individual outcomes, the panel agreed that the sum of these effects resulted in a small beneficial effect, and that the undesirable effects were trivial, except for the potential burden of the interventions. The panel agreed that the balance of effects between benefits and harms of SMIs vs. UC was, therefore, probably favourable for SMIs (low to very low certainty in the evidence of effects). Table 3 shows a summary of all the findings (see Appendix A).

A meta-regression of all SMIs vs. UC for the outcome HbA1c reduction, showed a reduction in patients with higher baseline risk; thus, SMIs may be more effective in patients with poor glycemic control (Appendix A). Subgroup analysis by intensity showed that for most outcomes “high intensity” interventions (offering more than ten contact hours) are associated with a larger improvement in the effects (for most of the outcomes) (Appendix A). Moreover, subgroup analyses included the presence vs. absence of each of the components. These revealed that when interventions did not include “education” or “monitoring” components, they tended to produce smaller effects. Subgroup analyses of other components showed a similar trend, but with a less clear pattern. However, these findings should still be interpreted with caution, as these effects may or may not be affected by the effect of other components (Appendix A).

We conducted heat plots by outcome as additional exploratory analyses to evaluate the effect of the different components and cluster of components [51]. The heat plot allows us to explore the behaviour of the components, by simultaneously visualising the effects of the two-by-two component combinations (Appendix A).

#### Values and Preferences

The evidence showed an important variability in how patients with T2DM value critical outcomes, mainly influenced by contextual factors and disease progression, among other factors (low to high certainty) [30]. Self-management is possible when patients can make adjustments and accept their diagnosis and treatment. Knowledge provision is better received in a positive patient–provider relationship. The decision-making process with healthcare providers enables engagement in SMIs enhancing patients’ self-efficacy. However, self-management achievement can be challenging, since it requires building capabilities, behavioural skills, social support, and attending scheduled care. For most patients and their caregivers, it is hard to perceive the risk of long-term complications. Being able to perform self-monitoring (as a component of a given SMI) facilitates awareness of glycaemic complications and glucose control. However, fear of hypoglycaemia and weight change may hinder adherence to treatment. The loss of quality of life, physical and psychological constraints, associated costs, comorbidities, and misleading life expectancy beliefs represent barriers to adhering to SMIs. Thus, to integrate self-management into everyday life, patients and caregivers require self-paced training according to their contextual factors.

#### Resources Required and Cost-Effectiveness

An increasing number of cost studies on SMIs in T2DM have been published over the last years. Thus far, only one SR has been published providing an overview of the available evidence of 12 studies [52]. This review found that costs for SMIs vary strongly between interventions, depending both on the design of the intervention and the costing perspective (i.e., which cost items are considered). A search of the COMPAR-EU database for costing studies and economic evaluations resulted in four additional studies providing evidence on resource use and costs. These 16 studies covered a wide variety of SMIs, and consequently, presented a wide range of cost estimates, ranging from EUR 4 to EUR 3747 (all costs determined in 2020). The mean per patient cost of an SMI was EUR 846, while the median cost was EUR 324.

The reduction in HbA1c, BMI, HDL, LDL, and systolic blood pressure as shown above led to a reduction in complication incidence and mortality, when applied in the UKPDS-OM2. This translated into a small incremental gain of 0.006 QALYs in the SMI group, compared to the current care group (Table 4). Direct medical costs (for the treatment of T2DM and related complications) was also lower in the SMI group. However, due to the increase in life years, other costs (e.g., informal care, unrelated medical costs) were higher in this group. Overall, the total costs (excluding the costs for the SMI) were slightly higher in the SMI group (EUR 67 per patient). In case of a willingness to pay of EUR 20,000 per QALY, an SMI producing these effects may cost up to EUR 218 per patient to be considered cost-effective.

SMIs are a very diverse type of intervention in terms of their structure, intensity, and resource use, amongst other things. Some interventions could be considered as having almost no resource requirements, while others are very resource intensive. As such, it may not be very informative to judge the resource use nor cost-effectiveness for ‘Any SMI’, as any particular SMI is likely to deviate substantially from the mean estimate. Likewise, interventions differ vastly in their effectiveness, and therefore, also their cost-effectiveness. Given this scenario, the panel agreed the cost-effectiveness does not favour either SMI or UC.

#### Equity

The panel agreed that if implemented tailored to culture and health literacy, it could increase equity. The use of peers facilitates tailoring, contributing to increasing equity. Equity might be affected by geographical location and accessibility, and by access to mobile technology.

#### Acceptability and Feasibility

The complexity of managing insulin has been associated with practical barriers that may lead to insulin non-adherence. Examples that are given are injection difficulties, patients failing to remember whether they have taken their insulin, difficulties in sustaining regular self-monitoring of blood glucose and adjusting insulin due to its additional burden and painfulness, regimen inflexibility, and the inconvenience of carrying an insulin pen with them and injecting regularly [53,54]. Specifically, comorbid conditions, when patients have to take multiple medications, represent a barrier to medication administration [55]. In addition, according to healthcare professionals, patients lack sufficient understanding of diabetes and knowledge in relation to insulin titration and dose adjustment, hampering the effective use of insulin [53]. The guideline panel judged that SMI are probably acceptable to key stakeholders. For patients and or caregivers, SMIs are acceptable overall, but may vary since it could be influenced by setting, accessibility, tailoring, and other factors. The guideline panel judged that overall, all SMI are probably feasible. Lack of human resources may make the implementation of some of them difficult.

#### 4.1.2. Justification of Recommendation

The COMPAR-EU T2DM panel made a conditional recommendation in favour of SMIs, rather than UC, due to a probably favourable balance of effects. The panel agreed that resource requirements and cost-effectiveness relationships vary, and that interventions probably increase equity and are probably acceptable and feasible. SMIs should be of high intensity (more than ten hours) and include education and monitoring plus one or more additional components, such as behavioural techniques (either action-based or emotional-based) and/or social support. Patients with T2DM with poor glycaemic control (poor HbA1c at baseline) are likely to experience a larger effect.

### 4.2. Recommendations for Selected SMI vs. UC

In patients with type-2 diabetes, the COMPAR-EU T2DM panel is in favour of using the following ten selected SMIs (see Table 4), rather than UC (conditional recommendation based on low certainty in the evidence of effects ⨁⨁◯◯).

#### 4.2.1. Summary of the Evidence

#### Evidence of Effects

A total of 24 NMAs were conducted for the different selected outcomes. Recommendations were informed with 13 NMAs (critical outcomes) and 2 standard meta-analyses for long-term complications and mortality. The NMAs vary in size, including from 11 to 463 trials. Analyses of the RCTs showed that the selected SMIs have from small to large desirable effects in at least one critical outcome (Table 5). For all selected SMIs, undesirable effects were considered trivial since no harmful effects were reported, except for the potential burden of the interventions. The panel agreed that the balance of effects between benefits and harms of SMIs vs. UC was favourable for SMIs, compared to UC (low certainty in the evidence of effects ⨁⨁◯◯).

All selected SMIs included an educational component. Most of the selected SMIs include monitoring techniques and/or behavioural techniques. A social support component is also present in several selected SMIs. Although most of the selected SMIs include a face-to-face component as part of the intervention, several interventions were delivered remotely. Most of the SMIs were delivered by professionals, although some could include the participation of peers (see Appendix A).

#### Values and Preferences

Although according to the literature there is variability on how individual patients value some critical outcomes, such as long-term complications, hypoglycaemia, and weight change, the panel judged that there was probably no important variability on how patients would value the outcomes when facing the decision to use the selected SMIs. Since in all selected SMIs, the balance of effects favours SMI (with at least a small beneficial effect outweighing the trivial undesirable effects), the panel agreed that most patients would choose the intervention.

#### Resources Required and Cost-Effectiveness

For the majority of the interventions selected, the panel judged that the resources required were moderate and that the cost-effectiveness does not favour either the intervention or the comparison. An intervention cost, given its effectiveness, and a threshold for cost-effectiveness varied across diseases and countries. Headrooms were estimated to range from EUR 0 to EUR 223, and from EUR 218 to EUR 8031, at a threshold of EUR 20,000 and EUR 50,000, respectively. Negative headroom indicates that SMIs do not have the potential to be cost-effective at the given threshold.

Headrooms also varied across countries, as the QALY gain per participant was different across countries due to the country specific input (e.g., risk factors at baseline and all-cause mortality). In general, the estimated headrooms were lower in Germany and the UK, and higher in Greece and the Netherlands. Because the short-term effects of SMIs were relatively small, assumptions regarding the duration of SMI effects were crucial and had a relatively large impact on the cost-effectiveness results, as shown in different scenario analyses.

The budget impact analyses showed the increase in costs in a specific country when the SMI would be implemented in the total target population based on the maximum costs at which the intervention could be considered cost-effective, as calculated in the headroom analyses. The results of the budget impact analysis, therefore, strongly depended on the headroom of SMIs for a specific disease, but also on the prevalence of the disease in a specific country.

#### Equity

The panel agreed that if SMI were implemented tailored to culture and health literacy, they could increase equity. The use of peers facilitates tailoring, thus contributing to increasing equity. Equity might be affected by geographical location and accessibility, and by access to mobile technology.

#### Acceptability and Feasibility

The panel judged that all SMIs are probably acceptable and feasible to key stakeholders. For SMIs including peers, the panel highlighted that these are probably more feasible in organisations or settings with a tradition of engaging peers in these types of interventions. Lack of access to human resources may make the implementation difficult.

#### 4.2.2. Justification of Recommendations

The panel formulated recommendations after discussing thoroughly each body of evidence summarised in the EtD frameworks (see Appendix A). The COMPAR-EU T2DM panel made conditional recommendations in favour of all selected SMIs, rather than UC, due to a probably favourable balance of effects. The panel agreed that for most interventions, resource requirements are moderate, that cost-effectiveness does not favour either the intervention or UC, and that interventions probably increase equity, and are probably acceptable and feasible.

As per the recommendation made for all SMIs vs. UC, these SMIs should be of high intensity (more than ten hours). Furthermore, patients with T2DM with poor glycaemic control (poor glycated haemoglobin at baseline) are likely to have a larger benefit than those with a better glycaemic control.

## 5. Discussion

### 5.1. Main Findings

The T2DM COMPAR-EU panel formulated an overall recommendation, suggesting in favour of using SMIs, rather than UC alone (conditional, very low certainty of the evidence about the effects). The justification for this recommendation is that the balance of the desirable and undesirable effects probably favours SMIs, based on low to very low certainty of the evidence of effects. SMIs should be of high intensity (ten hours or more) and include education, monitoring, behavioural techniques (either action-based or emotional-based), and/or social support.

We used the COMPAR-EU taxonomy to identify the components of each SMIs and grouped and defined each intervention as a combination of components (according to type of support technique, type of recipient, type of provider, and delivery methods). In total, we classified SMIs into 100 combinations of components that compose each of the NMA nodes. The panel formulated specific recommendations, suggesting the use of ten selected SMIs, rather than UC (conditional, low to very low certainty of the evidence about the effects). These recommendations were based on a probably favourable balance between benefits and harms on critical outcomes after reviewing the evidence of effects from the NMA and CNMA. The most frequent components across selected SMIs were education, monitoring, and behavioural techniques (either action-based or emotional-based).

Across recommendations, resource requirements were considered moderate, and cost-effectiveness analyses do not favour either the intervention or UC. However, there is a potential for SMIs to be cost-effective from a societal perspective, but this largely depends on the actual costs of providing these SMIs. The panel considered that SMIs probably increase equity, and that they are probably acceptable and feasible. The feasibility and fidelity of the interventions should also be assessed before implementation, and that SMIs should be tailored to populations with different levels of health literacy. These recommendations and their related decision aid tools are all included in a web-based platform (link to COMPAR-EU platform).

### 5.2. Our Results in Context of Previous Research

Current American Diabetes Association (ADA) standards of care include recommendations to implement several strategies, including education, telehealth delivery of care, digital solutions to deliver education and support, and the use of behavioural techniques, to support diabetes self-management and engagement in health behaviours to promote optimal diabetes health outcomes [56]. Likewise, the ESC/EASD Guidelines, include recommendations for group-based structured education programmes in patients with DM, to improve knowledge, glycaemic control, disease management, and patient empowerment, to facilitate shared control and decision making, to enhance self-efficacy, self-care, and motivation [46,57]. Our recommendations are consistent with these recommendations; however, our recommendations are conditional, rather than strong, given the rigorous assessment of the quality of evidence conducted, and the explicit incorporation of relevant criteria, such as values and preferences of patients and economic considerations. Furthermore, using novel statistical analyses, we explored the effect of each of the SMIs’ components and, thus, we were able to identify the most effective componentes, and the most promising potential combinations of components.

A previous SR with NMA reported similar findings to our review, showing that lifestyle and diabetes self-management education provided were beneficial especially when offered for 11 or more hours of contact; and that behavioural programs may improve glycemic control observing a greater reduction in patients with poor glycaemic control at baseline (i.e., baseline HbA1c level of 7.0% or greater) [58]. Related to implementation considerations, this review also stated that it would be beneficial to tailor programs to ethnic minority groups and to older adults, and to combine group interactions with peers [58]; likewise, our panel emphasised that SMIs should always be tailored to patients characteristics and context (more detailed information in the EtDs). Another SR focused on remote self-monitoring of blood glucose in T2DM patients, also showed that SMIs slightly improve glycaemic control and that the largest improvements were observed among patients with poor baseline HbA1c level 8% or greater. Furthermore, consistent with the reflections from the panel, the authors highlighted the need for higher quality primary studies that also address other outcomes, such as quality of life, quality of care, and cost-effectiveness to draw definite conclusions [59].

Cost-effectiveness studies of SMIs in T2DM from a societal perspective over a lifetime horizon have been scarce. To our knowledge, our analysis is the first to include informal care, productivity losses, and unrelated future medical costs. Therefore, the comparability of our results to previous research is limited. Previous modelling studies taking a lifetime horizon and evaluating a specific SMI concluded that it was either unlikely to be cost-effective, or that results were very uncertain [52], which is in line with our findings. These studies also noted that considerable uncertainty arose from lack of evidence on the duration of SMI effects and the need to make assumptions on this critical parameter, indicating that this is a key area for future research.

### 5.3. Limitations and Strengths

Our recommendations do not cover all the challenges of implementing SMIs into daily practice, due to limitations of the available literature (e.g., costs of implementation not reported, poor description of the interventions, and lack of information on the long-term outcomes). Thus, some of the gaps in the evidence identified in previous reviews remain, and it is still difficult to conclude the real clinical effect and impact of SMIs on health outcomes [57,60]. Almost all trials included in our review showed moderate or high risk of bias (mainly due to lack of blinding of participants, providers and researchers, and attrition bias). This is a key aspect that lowered our confidence in the evidence of effects and drove the panel decision to formulate conditional, rather than strong recommendations.

Details about the type of usual care were reported inconsistently. Not detailing the usual care presumes all control participants receive a similar standard of care within and across sites, and that usual care practices remain unchanged during the trial [61]. This may raise questions regarding what was included in usual care, whether usual care already had some self-management strategies in place, and how changes in usual care may affect the hypothesised outcomes. Without this information, interpreting the effectiveness of the intervention is challenging. Due to health systems varying widely, usual care can also be different, potentially influencing the results obtained from the different types of tested SMIs [62]. This is especially important for diabetes, considering that it is one of the diseases where behavioural programs are more established in practice.

Our NMA has several limitations, mainly because the network is sparse with most SMIs being compared to UC, and consequently, treatment effects are imprecise. Since most effect estimates are informed by direct evidence, the effectiveness reported in the studies was likely confounded by study characteristics. In the NMA, there was large statistical heterogeneity, a clear indication of clinical heterogeneity both in components and in usual care; subgroup analyses were exploratory. Furthermore, there is not a clear pattern (e.g., monotonic) between components and effectiveness. We also observed that the context in which a component is included is of paramount importance, which is consistent with other reviews showing that there is uncertainty on how other factors may impact on the effectiveness of an SMI [58].

Our recommendations also have several strengths. We followed a highly structured and explicit process, developing a suite of systematic reviews for the different types of evidence, applying the GRADE approach, including the use of evidence to decision frameworks. This is, to our knowledge, the largest SR on the effectiveness of SMI that analyses the effectiveness of each SMI, as a combination of different strategies or components. A well-known and validated health-economic model for T2DM was extended to adopt a societal perspective, and research on the societal impact of T2DM on productivity and informal care use in several European countries was conducted. The integration of the results of the systematic review on patients’ values and preferences and contextual factors is another strength.

### 5.4. Implications for Practice and Research

SMIs need to be discussed with patients and should be tailored to their characteristics and context. There are certain considerations that should be addressed when implementing SMIs, including contextual factors at healthcare providers’ level (it is important to adapt the advice, communication, or intervention to the patient’s personal situation and level of knowledge; to have adequate communication skills, for example, show empathy, provide understandable information, and ask questions), at patients’ level (their motivation to engage in self-management and patient’s attitude towards self-management, for example, beliefs about the importance of self-management for health and beliefs about the usefulness of certain self-management tasks), and at interaction level (patients’ preference regarding their own role in treatment, for example, the extent to which a patient wants to be involved in shared decision making and the extent to which a patient expects or wants professional involvement in the daily management of their disease).

Other specific implementation considerations for selected SMIs are described in detail in Appendix A (see Appendix A). Online decision aids have been developed within COMPAR-EU to guide these encounters. Guideline developers and policymakers may use these recommendations to adopt or adapt them to their settings.

Future studies should measure patient important outcomes, both critical and important (see Appendix A), have adequate sample size and follow-up. The feasibility and fidelity of implementation of the interventions should also be assessed. Populations should include patients with different levels of health literacy. The panel highlighted the importance of the need of conducting more rigorous research on these complex interventions, because of the lack of information around key aspects, such as the description of the usual care definition of the intervention, included components, duration, dose, and delivery method.

## 6. Conclusions

The COMPAR-EU panel formulated conditional recommendations in favour of using SMIs, rather than usual care, in T2DM according to the GRADE approach. Recommendations were based on evidence from systematic reviews of effects, values and preferences, contextual analysis, and cost-effectiveness analysis. Clinicians should engage patients in tailored discussions to guide them in making decisions aligned with their values, preferences, motivation, and attitudes toward self-management. Implementation considerations include addressing contextual factors at the healthcare provider’s level, such as adapting advice and communication to the patient’s situation.

## Figures and Tables

**Table 1 healthcare-12-00483-t001:** Structured clinical question.

Population	Intervention	Comparison	Critical Outcomes
Adult patients (>18 years) with T2DM	SMIs	Usual care (UC) ^1^ Any SMIs	HbA1c Weight management Quality of life and psychological distress Hypoglycaemia Blood pressure Lipid profile Long-term complications incidence Mortality incidence

^1^ For the formulation of recommendations, we only presented UC results; however, all comparators were included in the analyses.

**Table 2 healthcare-12-00483-t002:** Examples of self-management interventions defined as *“Educational, action-based behavioural techniques, delivered to patients individually, by health professionals”* using the COMPAR-EU taxonomy.

	COMPAR-EU Taxonomy Components				
**SMIs’ Examples**	**Type of Support Technique**		**Type of Recipient**	**Type of Provider**	**Delivery Methods**
Exercise programme where trainers tailored routines according to patients’ inputs	Educational (sharing information and skills training)	Action-based behavioural techniques (goal setting and develop action planning)	Individual	Physio- therapist	27 face-to- face sessions
Carbohydrate gram counting training using online resources, with individualised carbohydrate gram goals [34]	Educational (sharing information including previously designed material)	Action-based behavioural techniques (goal setting and develop action planning)	Individual	Dietician-certified diabetes educator	3 face-to-face sessions

**Table 3 healthcare-12-00483-t003:** Summary of findings for all SMIs vs. UC.

Outcome	Number of Participants (Studies)	Anticipated Absolute Effects (95% CI) Difference	Certainty	Plain Language Statement
**Glycated haemoglobin**	58,621 (335 RCTs)	**MD 0.39 % lower** (0.45 lower to 0.34 lower)	⨁◯◯◯**Very low** ^a^	SMIs may have a **marginal beneficial effect** ^b^ on glycated haemoglobin, but the **evidence is very uncertain**
**Long-term complications inferred from blood pressure:**				
**Systolic blood pressure**	31,526 (173 RCTs)	**MD 4.29 mmHg lower** (2.11 lower to 1.49 lower)	⨁⨁◯◯ **Low** ^c^	SMIs may have a **marginal beneficial effect** ^d^ on systolic blood pressure
**Diastolic blood pressure**	29,047 (154 RCTs)	**MD 1.12 mmHg lower** (1.53 lower to 0.73 lower)	⨁⨁◯◯ **Low** ^e^	SMIs may have a **marginal beneficial effect** ^f^ on diastolic blood pressure
**Long-term complications inferred from lipid profile:**				
**Triglycerides**	17,462 (128 RCTs)	**MD 0.12 mmol/L lower** (0.20 lower to 0.03 lower)	⨁◯◯◯ **Very Low** ^g^	SMIs **may** have a **marginal beneficial effect** ^h^ on triglycerides, but the **evidence is very uncertain**
**Low-density lipoprotein**	21,140 (123 RCTs)	**MD 1.91 mg/dL lower** (3.29 lower to 0.53 lower)	⨁◯◯◯ **Very Low** ^i^	SMIs **may** have a **marginal beneficial effect** ^j^ on low-density lipoprotein, but the **evidence is very uncertain**
**Weight management:**				
**Body Mass Index**	29,494 (174 RCTs)	**MD 0.26 kg/m^2^ lower** (0.41 lower to 0.12 lower)	⨁◯◯◯ **Very Low** ^k^	SMIs **may** have a **marginal beneficial effect** ^l^ on body mass index, but the **evidence is very uncertain**
**Waist size**	9500 (65 RCTs)	**MD 1.23 cm lower** (1.92 lower to 0.66 lower)	⨁⨁◯◯ **Low** ^m^	SMIs **may** have a **marginal beneficial effect** ^n^ on waist size reduction
**Weight**	16,124 (96 RCTs)	**MD 0.89 Kg lower** (1.37 lower to 0.42 lower)	⨁⨁◯◯ **Low** ^o^	SMIs **may** have a **marginal beneficial effect** ^p^ on weight management
**Quality of Life**	10,169 (63 RCTs)	**SMD 0.18 SD higher** (0.03 higher to 0.34 higher)	⨁⨁◯◯ **Low** ^q^	SMIs **may** have a **marginal beneficial effect** ^r^ on quality of life
**Psychological distress**	5481 (26 RCTs)	**SMD 0.31 SD lower** (1.17 lower to 0.55 higher)	⨁⨁◯◯ **Low** ^s^	SMIs may have a **marginal beneficial effect** ^t^ on psychological distress
**Hypoglycaemia**	1788 (6 RCTs)	**Rate Ratio 0.89** (0.56 to 1.42)	⨁◯◯◯ **Very Low** ^u^	SMIs **may** have a **marginal beneficial effect** ^v^ on hypoglycaemia events, but the **evidence is very uncertain**

CI: confidence interval; MD: mean difference; SMD: standardised mean difference; RCT: randomised control trial; SMI: self-management intervention; Low certainty (⨁⨁◯◯): our confidence in the effect estimate is limited; the true effect may be substantially different from the estimate of the effect; Very low certainty (⨁◯◯◯): we have very little confidence in the effect estimate; the true effect is likely to be substantially different from the estimate of effect. **Explanations** a. Downgraded due to serious risk of bias and very serious inconsistency (Tau2 = 1.839; I^2^ = 0.9899). b. The effect was below the minimal important difference (MID) threshold of 0.5. c. Downgraded due to serious risk of bias and serious inconsistency (Tau2 = 9.0348; I^2^ = 0.8212). d. The effect was below the MID threshold of 5 mmHg. e. Downgraded due to serious risk of bias and serious inconsistency (Tau2 = 3.4373; I^2^ = 0.8765). f. The effect was below the MID threshold of 2.5 mmHg. g. Downgraded due to serious risk of bias and very serious inconsistency (Tau2 = 0.1403; I^2^ = 0.9007). h. The effect was below the MID threshold of 2 mmol/L. i. Downgraded due to serious risk of bias and very serious inconsistency (Tau2 = 35.1182; I^2^ = 0.9165). j. The effect was below the MID threshold of 38.7 mg/dL. k. Downgraded due to serious risk of bias and very serious inconsistency (Tau2 = 0.5091; I^2^ = 0.919). l. The effect was below the MID threshold of 2.5 kg/m^2^. m. Downgraded due to serious risk of bias and serious inconsistency (Tau2 = 2.4503; I^2^ = 0.6503). n. The effect was below the MID threshold of 8 cm. o. Downgraded due to serious risk of bias and serious inconsistency (Tau2 = 2.1513; I^2^ = 0.7322). p. The effect was below the MID threshold of 11.6 kg. q. Downgraded due to serious risk of bias and serious inconsistency (Tau2 = 0.3343; I^2^ = 0.8759). r. The effect was below the MID threshold of 0.5 SD. s. Downgraded due to serious risk of bias and serious inconsistency (Tau2 = 0.0437; I^2^ = 0.6927). t. The effect was below the MID threshold of 0.5 SD. u. Downgraded due to serious risk of bias, serious impression, and serious inconsistency (Tau2 = 0.209; I^2^ = 0.7988). v. The effect was below the MID threshold of 13 events per 1000 patient-year.

**Table 4 healthcare-12-00483-t004:** Cost-effectiveness (all SMIs vs. UC, UK population).

Outcomes	Mean Difference	Certainty	Plain Language Statement
QALYs (1 model; lifetime)	**0.006 more QALYs per person** (0.013 fewer to 0.034 more)	⨁⨁◯◯ **Low** ^a,b^	The intervention may have little to no effect on QALYs
Total costs without treatment costs (EUR 2020) (1 model; lifetime)	**67 EUR more per person** (839 fewer to 994 more)	⨁◯◯◯ **Very Low**^b^	The evidence on the incremental cost per person is uncertain
Headroom (1 model; lifetime)	**218 EUR** (0 to 1360)	⨁⨁◯◯ **Low**^a,b^	The intervention may cost up to 218 EUR per person and still be considered cost-effective ^c^

NA: non-applicable; QALY: quality adjusted life years; ICER: incremental cost-effectiveness ratio; GBP: Great British Pound; EUR: euro. **Explanations** a. Risk of bias: we downgraded two levels due to a low to very low certainty of evidence on the input parameters for the model. b. The model used the following input parameters from the NMA: HbA1c, BMI, systolic blood pressure, HDL cholesterol, and LDL cholesterol. c. For the headroom analysis a threshold of 20,000 EUR per QALY was used to consider an intervention to be cost-effective.

**Table 5 healthcare-12-00483-t005:** Summary of findings of selected SMIs (effect and certainty).

SMIs for T2DM	Large Beneficial	Moderate Beneficial	Small Beneficial	Marginal to No Effect
**Monitoring techniques lead by peers delivered in groups**	It may decrease systolic blood pressure levels, but the evidence is very uncertain	It may decrease HbA1c levels	-	It may result in little to no effect on waist size, body mass index, and diastolic blood pressure, but the evidence is very uncertain
⨁⨁◯◯ **Low**^a^	⨁◯◯◯ **Very Low** ^b,d^	⨁⨁◯◯ **Low** ^c,d^	-	⨁◯◯◯ **Very Low** ^b,d^
**Emotional-based behavioural techniques lead by peers delivered remotely**	It may decrease HbA1c levels	-	It may result in a slight decrease in hypoglycaemic events	-
⨁⨁◯◯ **Low** ^a^	⨁⨁◯◯ **Low** ^c,d^	-	⨁⨁◯◯ **Low** ^c,d^	-
**Monitoring and action-based behavioural techniques and shared decision making, and social support delivered in groups**	-	-	It may result in a slight decrease in HbA1c levels	-
⨁⨁◯◯ **Low** ^a^	-	-	⨁⨁◯◯ **Low** ^c,d^	-
**Monitoring, action-based and emotional-based behavioural techniques, and social support led by peers delivered remotely**	-	It may decrease diastolic blood pressure	It may result in a slight decrease in HbA1c levels and psychological distress, but the evidence is very uncertain	It may result in little to no effect on systolic blood pressure and weight reduction
⨁⨁◯◯ **Low** ^a^		⨁⨁◯◯ **Low** ^c,d^	⨁◯◯◯ **Very Low**	⨁⨁◯◯ **Low** ^c,d^
**Emotional-based behavioural techniques and social support delivered in groups**	It may decrease HbA1clevels	-	-	It may result in little to no effect on triglycerides and LDL
⨁⨁◯◯ **Low** ^a^	⨁⨁◯◯ **Low** ^c,d^	-	-	⨁⨁◯◯ **Low** ^c,d^
**Action-based behavioural techniques, social support led by peers and professionals**	It may decrease HbA1c levels	-	-	It may result in little to no effect on BMI, but the evidence is very uncertain
⨁⨁◯◯ **Low** ^a^	⨁⨁◯◯ **Low** ^c,d^	-	-	⨁◯◯◯ **Very Low** ^c,e^
**Education delivered in groups and remotely**	-	It may decrease HbA1c levels	-	-
⨁⨁◯◯ **Low** ^a^		⨁⨁◯◯ **Low** ^c,d^	-	
**Monitoring techniques and social support delivered remotely**	-	-	It may result in a slight decrease in HbA1c levels	It may result in little to no effect on BMI, weight, triglycerides, and LDL ⨁⨁◯◯ **Low** ^c,d^
⨁⨁◯◯ **Low** ^a^			⨁⨁◯◯ **Low** ^c,d^	It may result in little to no effect on systolic blood pressure and diastolic blood pressure ⨁◯◯◯ **Very Low**
**Monitoring and action-based behavioural techniques, shared decision making and social support, delivered in groups**	-	-	It may result in a slight decrease in HbA1c levels	-
⨁⨁◯◯ **Low** ^a^			⨁⨁◯◯ **Low** ^c,d^	
**Monitoring and emotional-based behavioural techniques delivered remotely**	-	-	It may result in a slight decrease in HbA1c levels	-
⨁⨁◯◯ **Low** ^a^			⨁⨁◯◯ **Low** ^c,d^	

BMI: body mass index; DBP: diastolic blood pressure; HbA1c: glycosylated haemoglobin; LDL: low-density lipoprotein; QoL: quality of life; SBP: systolic blood pressure; SMIs: Self-management interventions; T2DM: type 2 diabetes mellitus; Low certainty (⨁⨁◯◯) our confidence in the effect estimate is limited; the true effect may be substantially different from the estimate of the effect; Very low certainty (⨁◯◯◯): we have very little confidence in the effect estimate; the true effect is likely to be substantially different from the estimate of effect. Explanations: a. The guideline panel judged the overall certainty of evidence as low, being HbA1c the most relevant outcome and the certainty of that evidence being low. b. Downgraded due to very serious risk of bias. c. Downgraded due to serious risk of bias. d. Downgraded due to serious imprecision. e. Downgraded due to very serious imprecision.

## Data Availability

Data are contained within the article and Appendix A.

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
