# Peer review of "COMPAR-EU Recommendations on Self-Management Interventions in Type 2 Diabetes Mellitus"

_healthcare, 2024, doi:10.3390/healthcare12040483_

Round 1

Reviewer 1 Report

Comments and Suggestions for Authors

The manuscript is interesting and well-composed. The aim of this work is to provide evidence-based clinical recommendations, for the use and implementation of SMIs in adults with T2DM. The authors exploited the COMPAR-EU project, a multimethod, interdisciplinary project which aims to bridge the gap between current knowledge and practice of SMIs for adults in Europe living with one of the four high-priority chronic conditions including T2DM. 

Here below some suggestions: 

- it is necessary a moderate editing of English language, expecially in the abstract, because some paragraphs are difficult to be understood
- Please explain in the abstract and in the introducton briefly what is the GRADE approach 
- what are the main caracteristics of the study populations considered in this analysis? 

Comments on the Quality of English Language

The manuscript is interesting and well-composed. The aim of this work is to provide evidence-based clinical recommendations, for the use and implementation of SMIs in adults with T2DM. The manuscript is well explained. However, I suggest minor revision of English language in the abstract and introduction paragraphs. 

After these minor revisions, I think that the manuscript is suitable for pubblication. 

Author Response

Dear Reviewer,

We would like to thank you for the valuable comments to improve our manuscript “COMPAR-EU Recommendations on Self-management Interventions in Type 2 Diabetes Mellitus” ID healthcare-2708802, submitted for publication in Healthcare Journal. Below, we include a point-by-point response to your comments and suggestions, as well as the quotation of the text changes, referring to the line numbers in the attached version.

  • Moderate Editing of English Language:

 Thank you very much for your suggestion to moderate editing of the English language. After a thorough review of the English language, particularly in the abstract, we performed a comprehensive editing process to enhance the clarity of paragraphs and ensure that the content is easily understood. Please find the corresponding modifications of the English language in the abstract in the new submitted version. 

  • Explanation of GRADE Approach in Abstract and Introduction::

We understand the importance of providing a concise explanation of the GRADE approach in both the abstract and the introduction. In response to your suggestion, we have added a brief explanation of the GRADE approach in the abstract (lines 6-8) and the introduction (lines 70-73 ).

  • Characteristics of Study Populations:

We appreciate your interest in understanding the main characteristics of the study populations considered in our analysis. The descriptive analyses of the characteristics of the studies included in the systematic review, such as study population characteristics, were the scope of another publication of the COMPAR-EU project. To address your concerns, we provide a brief explanation below, and we have cited this article in our manuscript (lines 103-108).

We are committed to incorporating these revisions to enhance the overall quality and clarity of our manuscript. Your constructive feedback has been invaluable, and we are confident that these adjustments will significantly improve the reader's experience.

Thank you once again for your thoughtful review,

Sincerely,

Jessica Beltran

Researcher- Cochrane Iberoamerican Centre

Reviewer 2 Report

Comments and Suggestions for Authors

The paper is supposedly intended to report the results of part of a larger study, but it does not appear to be self-contained as a paper, forcing the reader to necessarily refer to the literature for knowledge of the study itself and understanding of the meaning of terms in order to understand its contents.

The content of the paper seems credible enough, with many experts contributing as co-authors.

However, the main conclusion, Recomendation for SMIs vs UC, is very low certainty in the evidence (with not so common pictograms), and individual recommendation for selected SMIs are also described as low, and difficult for the general reader to understand the objective level of the recommendation. The SMIs themselves should be explained in more detail in the introduction, and it is unclear to what extent the usual care (UC) used as control is proper (commonnly accepted) diabetes care. Elsewhere, methods and terminology in the papers that seem obvious to experts but difficult to understand for the general reader seem to be commonly used.

Certainly, the boxed commentary and the many references to the necessary literature seem to provide sufficient information, making the paper useful for specialists in this particular area, but it does not seem to be sufficiently friendly for the general reader.

It is strongly recommended that the text of this paper be made more compact and changed to be self-contained, i.e. understandable to the general reader without reference to other sources and all information that is not directly necessary to understand the conclusions should be moved to supplements and separated from the text.

Author Response

Dear Reviewer,

We would like to thank you for the valuable comments to improve our manuscript “COMPAR-EU Recommendations on Self-management Interventions in Type 2 Diabetes Mellitus” ID healthcare-2708802, submitted for publication in Healthcare Journal. 

Below, we include a point-by-point response to your comments and suggestions, as well as the quotation of the text changes, referring to the line numbers in the attached version.

Certainty of Evidence and Recommendation Clarity:

We acknowledge the issue regarding the low certainty of evidence and the difficulty for the general population in understanding the objective level of the recommendation. In response, we have provided a brief description of what a conditional recommendation and what low/very low certainty means (see line 44, Table 3 and Table 5).

Explanation of SMIs in the Introduction:

We recognize the importance of providing a more detailed explanation of SMIs in the introduction to aid comprehension for all readers. We have expanded and clarified our introduction, offering a comprehensive overview of the SMIs under consideration (See lines 59-61).

Clarity on Usual Care (UC):

We understand the need for clarity regarding the extent to which usual care (UC) used as control aligns with commonly accepted diabetes care. In the revised manuscript, we have provided a more explicit description of UC, ensuring that its components are well-defined and understood (see lines 119-122).

Accessibility for General Readers:

We appreciate your feedback on the paper's readability for general readers. To address this concern, we have made the text more concise, self-contained, and easily understandable. Information not essential for understanding the conclusions have been moved to supplements.

Compactness and Self-Containment:

To enhance readability and accessibility, we have worked on making the text more compact and self-contained.  However, please consider that this manuscript is only a part of a larger project, and Box 1 inclusion was agreed with Journal editors. Supplementary materials will be appropriately utilised to provide detailed information without burdening the main text.

We genuinely value your feedback and are committed to ensuring that our article is both informative for specialists and accessible to a broader readership. We believe these revisions will significantly improve the overall quality and readability of the manuscript.

Thank you once again for your insightful comments.

Sincerely,

Jessica Beltran

Researcher- Cochrane Iberoamerican Centre

Reviewer 3 Report

Comments and Suggestions for Authors

The study described the recommendations of the COMPAR-EU panel on using self-management interventions  (SMIs) over usual care (UC) for Type 2 Diabetes patients. The study also provided an excellent summary of the individual methodological components of SMIs with examples and their beneficial effects on type 2 diabetes subjects. The study further provided evidence of the beneficial effect of the cumulative and individual components of SMI on different outcomes of T2D subjects. The study also provided detailed considerations for patients and healthcare providers that need to be considered while implementing SMIs. The study also detailed the differences in the cost of resources used for SMIs across European countries and compared the cost-effectiveness of SMI components with Usual care. The authors presented the conclusion of the COMPAR-EU panel evaluation that for patients with T2DM, the SMIs will be more beneficial when compared with usual care (UC) alone.  

Authors are advised to pay attention to minor details, such as providing full forms for abbreviations used in the manuscript. Example: RCT, COPD.

Also, RCT is misspelled as “RTC” in lane 241.

Author Response

Dear Reviewer,

Thank you for your constructive feedback on our  manuscript “COMPAR-EU Recommendations on Self-management Interventions in Type 2 Diabetes Mellitus” ID healthcare-2708802, submitted for publication in Healthcare Journal. We appreciate your positive comments and valuable insights. 

We have carefully considered your suggestions and made the necessary revisions to enhance the quality of the manuscript. Here are the specific actions we have taken:

Full Forms for Abbreviations:

We acknowledge the importance of providing full forms for abbreviations used in the manuscript. In response, we have ensured that all abbreviations, including RCT and COPD, are accompanied by their full forms upon first mention in the text.

Correction of Spelling Error:

We appreciate your keen eye in identifying the misspelling of "RCT" as "RTC" in line 241. This typographical error has been rectified in the revised manuscript.

We believe that these revisions will contribute to the clarity and accuracy of the manuscript. We are grateful for your meticulous review, and we are confident that these adjustments will improve the overall quality of the article.

Thank you once again for your time and valuable feedback.

Sincerely,

Jessica Beltran

Researcher- Iberoamerican Cochrane Center

Round 2

Reviewer 2 Report

Comments and Suggestions for Authors

The manuscript has been subjected to the necessary improvements and is acceptable in this form if the authors do not wish for further improvements.